# Forward Modeling for Partial Observation Strategy Games - A StarCraft Defogger

## Abstract

This paper we present a *defogger*, a model that learns to predict future hidden information from partial observations. We formulate this model in the context of forward modeling and leverage spatial and sequential constraints and correlations via convolutional neural networks and long short-term memory networks, respectively. We evaluate our approach on a large dataset of human games of StarCraft: Brood War, a real-time strategy video game. Our models consistently beat strong rule-based baselines and qualitatively produce sensible future game states.

## 1 Introduction

We consider the problem of joint state estimation and next-state prediction in partially observable environments with complex dynamics. We take as a concrete example the problem of *defogging* in the real-time strategy (RTS) video game StarCraft, which we define as predicting the features of the game state that are hidden to the player.

Forward modeling, the prediction of what is going to happen next, is a core enabler both for reactive control and for longer term planning. Many researchers are attempting to build and create algorithms that are able to model the future, especially in next frame video prediction and robotic planning (Luc *et al.* , 2017; Agrawal *et al.* , 2016) One particular difficulty of forward modeling is to deal with the uncertainty of making a prediction with only a partial model and a partial view of the world. Jaakkola *et al.* (1995); Hostetler *et al.* (2012).

In RTS games such as StarCraft, players must build an economy and control agents, called units, on a 2 dimensional grid to overcome their opponents. Several inherent limitations of any real-world setting are made explicit in such RTS games. First, by the "fog of war" which only allows players to see the surroundings of their own units and are thus unable to fully access the true game state. Second, the low-level dynamics are extremely complex, because several hundreds of agents interact together. However, there is an implicit spatio-temporal structure that makes long-term reasoning depend mostly on lower-resolution abstractions that can be obtained by averaging fine-grained characteristics over time and space. This poses a challenge for both human and computer players alike and predicting hidden information is a key step towards efficient planning in such environments,

In this paper, as a first step towards learning a fully-featured forward model of the environment, the task we propose is to uncover hidden information and to predict the next state from observational data. We present a comprehensive analysis of a StarCraft *Defogger*, which predict features of the game at different levels of granularity: global features of the game such as the buildings of the opponent, and local features such as army density averaged by regions. Starting from a map of the environment subsampled in time and space, we propose a deep architecture of stacked long short-term memory cells applied convolutionally in an encoder-decoder architecture to predict the full state at different spatial resolutions. Individual layers of convolutional LSTMs encode the dynamics of the local features, and are aggregated in subsequent layers to model lower-resolution movements and global features. Trained on a large dataset of human replays (Lin *et al.* , 2017), the model significantly outperforms strong rule-based baselines on several metrics.

## 2 RELATED WORK

In this paper, we look at inferring the hidden information, from partial observations, either to make use of the inferred values directly, or to use them to decide what subsequent information to gather, i.e. lift our known unknowns. We do so with a forward model (see Section 4), because it allows us to leverage the spatio-temporal structure of the game.

StarCraft is an imperfect information game with a rich body of research (Ontanón *et al.* , 2013). Few works attempt to deal with the fog of war: by using particle filtering to estimate the position of the opponent units (Weber *et al.* , 2011), or by predicting where, how and when the opponent will attack (Synnaeve & Bessiere, 2012). Poker is another imperfect information game, on which research started jointly with the field of game theory. Moravčík *et al.* (2017); Brown & Sandholm (2017) yielded AIs that beat human professionals.

One can draw a distant parallel between the general problem of *defogging* and instance segmentation in vision (when in presence of occlusions) (Silberman *et al.* , 2014). Learning a forward model can be used to do video frame prediction (Mathieu *et al.* , 2015). Methods predicting in feature space include (Luc *et al.* , 2017), in which the authors predict future frames in the semantic segmentation space.

Forward models have a rich history in control theory, various parametric and nonparametric methods were proposed for optimal filtering of different classes of stochastic systems (Jazwinski, 2007; Crisan & Rozovskii, 2011). Attempts were made to treat partial observations cases (Krishnamurthy, 2016), but not in the case of branching processes. Oh *et al.* (2015) enhances the exploration of a deep reinforcement algorithm with a learned predictive model in Atari. Weber *et al.* (2017) learn a forward model (jointly with a model-free controller) and use it for planning in Sokoban and MiniPacman. Finally, Uriarte & Ontanón (2015) estimate combat outcomes in StarCraft by employing forward models in a Monte Carlo Tree Search algorithm.

## 3 "DEFOGGING": PROBLEM STATEMENT

"Defogging" consists of recovering the whole state $s_t$ from partial observations $o_{1...t}$ in imperfect information games. We explain why it matters in RTS games.

### 3.1 FORMALISM

Let $\{U\}^n$ denote the set of all sets $\{u_1, ..., u_n\}$ consisting of $n$ units, $u_i \in U$ with $U$ being unit types. As a state space we shall use $S = \bigcup_{n=1}^{N} \{U\}^n$ where $N$ is the maximum number of player's units. We will consider two-player games for which the full game state at time $t$ is $s_t = (s_t^{(1)}, s_t^{(2)}) \in S^2$. At each time-step $t$, each player $p$ receives observations of state $s_t$. We assume that each player fully observes her own state $s_t^{(p)}$ but only observes a subset of her opponent's units, corresponding to what she can see according to the rules of the game.

These visibility rules for player $p$ can be represented as a function $\zeta^{(p)} : S^2 \to S$ which maps a state $s_t$ to $\{v \in \{s_t^{(1)}, s_t^{(2)}\} \mid \exists u \in s_t^{(p)} \text{ s.t } u \text{ can see } v\}$.

The task of *defogging*, akin to filtering, consists in deriving the full state $s_t$ from the observations of one player $o_{1...T}^{(p)} = \{(s_t^{(p)}, \zeta^{(p)}(s_t)\}_{t=1}^{T}$. Hence, the partial observation setting considered here is different from the classical one, where only diffusion in the state space considered or the moments of branching are supposed to be known.

### 3.2 CHALLENGES AND IMPLICATIONS FOR RTS GAMES

In Real-Time Strategy games, the task of *defogging* poses several challenges. First, it requires remembering virtually everything that was shown even when those units are hidden under the fog of war again (long term memory), with as many characteristics as possible, as seeing a unit of a given type a given location at a given time may represent different information sets (or lead to different belief states) than if any of these characteristics was changed. Second, excellent performance in

*defogging* can only be achieved by a model that can reason about this memory and apply inference steps that correspond to the game dynamics. Third, such a model can leverage the correlations of observations (due to the distribution over valid strategies), but should not overfit those. Finally, in most RTS games, inference in such a model from raw state representation is a real computational burden. As a consequence, it is necessary to formulate abstractions for making model training and evaluation tractable.

Even though it is difficult to even estimate the set of valid states for $s_t$ with $o_{1...t}$ precisely, humans have no difficulty in obtaining a useful estimate of the state of the opponent. It allows them to rule out large subsets of the possible states, which would reduce the computational cost of any algorithmic approach (even for model-free control, it would lower the variability in the input). Also, their estimate of the state of the opponent allows them to efficiently gather information (known unknowns). Having a good enough *defogger* would allow to test direct perfect information approaches in real RTS settings. Finally, human professional players exhibit theory of mind (recursive, counterfactual, reasoning about both players strategies) based on their inferred state. Algorithmic *defoggers* may enable a more game theoretic approach to the abstract strategic view of RTS games.

## 4    "DEFOGGING" MODELS

We propose models based on an encoder-decoder architecture, as baselines for the *defogging* problem.

### 4.1    ENCODER-DECODER MODELS

We consider the formulated task as a parametric estimation problem. The goal is to reconstruct some coarse representation of $s_t$ from the observations $o_{1...t}$ and predict both the number of units $n$ in the opponent's state and their approximate states $\{u_1, ...., u_n\}$ that include unit type and location. We take two types of state representation in our approach: 1) unit counts per type and map region, and 2) unit type existence in a map region. The map is split into a grid of $H \times W$ cells. For each cell $(c_x, c_y)$, we compute $x^{(p)}_{t,c_x,c_y} \in \mathbb{N}^d$ which contains counts of units of player $p$ for each unit type, and $y^{(p)}_{t,c_x,c_y} \in \{0,1\}^d$ that accounts for unit type presence or absence in the cell. Here $d$ is the number of different unit types. To estimate parameters, we consider two types of loss functions: Huber ("smooth L1") loss $\mathcal{L}_r(\hat{x}, x)$ for unit type counts and binary cross-entropy loss $\mathcal{L}_c(\hat{y}, y)$ for unit presence. We denoted $x = (x^{(1)}_{t_1...t_2,1...W,1...H}, x^{(2)}_{t_1...t_2,1...W,1...H})$ and $y = (y^{(1)}_{t_1...t_2,1...W,1...H}, y^{(2)}_{t_1...t_2,1...W,1...H})$ to keep notations concise. As we strive for one model that reconstructs both representations, we train models with $\mathcal{L} = \mathcal{L}_r + \lambda \mathcal{L}_c$.

We restricted the family of models to explore by having an encoder-decoder approach (as shown in Fig. 1), in which we encode $x$ with a ConvNet ($E$), which does poolings in $c_x$ and $c_y$ coordinates if necessary so that we only have one vector per time-step. Then we apply a temporal inference mechanism and memory ($R$), for which we mostly experimented with recurrent neural networks. We tried using a simple temporal convolution, but did not notice significantly increased improvements in our metrics. We concatenate the localized embeddings of $x$ with the output of $R$, and we decode with a ConvNet ($D$). Finally we apply a regression head ($H_r$) and a classification head ($H_c$) to each of the decoded vectors in each coordinate. In practice we do 2D spatial pooling before the classification head $H_c$.

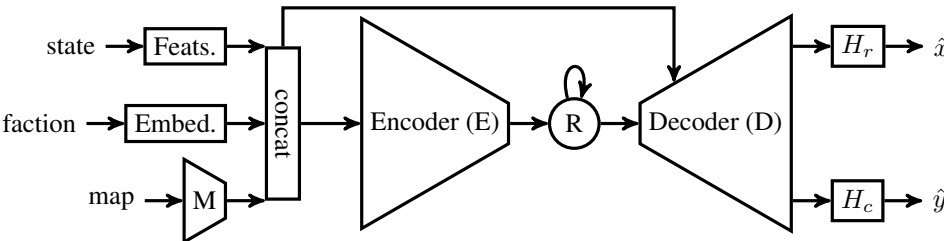

Figure 1: Simplified architecture of the model. Rectangles denote fully connected layers ($1 \times 1$ convolutions) or MLPs, trapezes denote convolutional neural networks, circles denote recurrent neural networks.

## 4.2 FEATURES

Because the individual statistics of each unit are difficult to predict on a macro level, and the resolution and dimensionality would be too high to be efficiently tractable, we downsample the state space into the counts of units per unit type, in evenly spaced blocks, at larger time steps than the game engine tick.

The dimensions of the most popular (and representative) map are $512 \times 512$ walktiles, we do different models with spatially sum-pooled tiles from $32 \times 32$ with a striding of 32, up to $128 \times 128$ with a striding of 64 walktiles. Precisely, when the game map has a true height and width of $H_0$ and $W_0$, we apply a *grid* of size $g$ to downsample it into a map of height $H = \frac{H_0}{g}$ and $W = \frac{W_0}{g}$ Our feature map of height $H$ and width $W$ is of size $T \times C \times H \times W$, where $T$ is time, $C$ has the first half reserved for our own units and the second for enemy units. To produce coarse frame predictions, we do not need to do so at a small time difference. However, skipping a large number of frames between each prediction runs into the problem where during a period of time, a unit may appear and disappear into the fog of war, as we uncover a part but cover it up soon after. If this quick glance of the enemy is not featurized, we lose key information during the scouting process. Thus, we combine frames instead of simply skipping, and featurize units by their last seen location in a set of frames. With a *step* of $s$ frames, we downsample the game in time to create $T = \frac{T_0}{s}$, with the real game taking $T_0$ frames.

The full game state holds auxiliary information that are essential to making good predictions in the future. The opponent faction (or race), if selected, is key to predicting the type of units that we need to predict. The map layout is key to predicting unit movement, as ground units cannot walk across cliffs, as well as give restrictions as to the starting locations (before any opponent units has been seen). We featurize the faction as a static embedding, replicated $H \times W$ times and concatenated to each time slice along the feature dimension We also have a ConvNet ($M$) that takes in features specific to each map: the walkability of each tile, buildability of each tile, ground height of each tile, and starting locations. This ConvNet is designed with the same grid size and stride as the featurizer, to ensure a $H \times W$ output size.

We consider several different models, and we have done a search through model space to find the most effective models for this problem. We employ random hyperparameter search followed by manually guided random plus grid searches to find the best hyperparameters for our models.

We did a preliminary search over grid sizes, and choose a grid of $32 \times 32$ predicting 15 seconds at a time to do most experiments on. We reason that 30 seconds is sufficient for some fast units to reach across the map, so it should be a good upper bound on how well we can expect a *defogger* to perform. We also present results for a grid of $64 \times 64$, as well as models with time skips of 5 and 30 seconds. Finally, we fix the dataset such that we predict minutes 3 to 11, with the reasoning that the very early game is not as interesting since enemies have not been found, and any prediction might just be guessing, and the most interesting parts of the openings happen during this time frame. We choose not to use later than 11 minutes to avoid the long tail, since variability increases with game duration.

## 4.3 MODEL DETAILS

We tried several different specifications of the model described in Fig. 1. We tried vanilla ConvNets, as well as one with enough *striding* such that the map is eventually downsampled into a $1 \times 1$ feature vector before $R$, thereby avoiding a pooling layer at the end of the encoder. We also try a *conv-lstm* encoder, where there are recurrent connections in the encoder. This is done by tiling the encoder recurrent layer one after the other $E_1 \rightarrow R_1 \rightarrow E_2 \rightarrow R_2 \rightarrow ....$, where each encoder does some downsampling and each recurrent layer is replicated across the spatial grid. The recurrent layer ($R$) was always an LSTM, except for a *conv-only* model in which we replaced it with a temporal convolution. The decoder ($D$) was always another ConvNet, whose properties where other hyperparameters, with the same activation function (or residuals) as the encoder, without LSTMs ever. Its padding was set at all layers to be half the kernel size of the convolution rounded down, so that it produces exaclty one output tile per input tile.

We did random search over the space of hyperparameters. We searched over two optimizers (SGD, Kingma & Ba (2014)), the regression loss (MSE, Huber), $\lambda$ values, and predicting the target value directly or predicting the difference between input and target, number of dimensions for hidden units (128, 256), number of layers (2,4,9). We also searched over using residual blocks, gated convolutions,

different nonlinearities.In our final models, we noticed several trends. Generally, gated convolutions Dauphin *et al.* (2017) did the best, while residual blocks with an information bottleneck did not improve over ReLU (probably due to the limited depth of our models). Adam was the more robust optimizer and converges faster than SGD, though SGD might converge to a better minimum given that it was trained for long enough.

We noticed that the most robust (across tasks) model was the conv-lstm. The simple model performed the worst, as too much information is lost in the pooling layer after $E$ (before $R$), and this layer is hard to avoid due to the difference in sizes of StarCraft maps. The striding model could learn, but is much more sensitive to hyperparameters, as was the conv-only model. Additionally, the lack of a memory in conv only make it much more difficult to reason about long term dependencies, as each frame of the model must reason about details such as the location of the enemy base, and therefore is less computationally efficient.

In reference to section 4.1 and Fig. 1, we report results with: $\mathcal{L}_r$ Huber, $\mathcal{L}_c$ binary cross entropy, $R$ LSTM, $D$ vanilla ConvNet, and two different flavors of $E$. For conv-lstm, we show results for a model with depth $d$, meaning that there are $d$ instances of a $d$-convolution layers followed by downsampling and a recurrent layer (so a $d = 2$ corresponds to 4 convolutions, $d = 3$ to 9). For striding, we show results for a model with depth 4, entailing that the input is downsampled by a factor of 16, so in most cases there is no global pooling at the end.

## 5 Experiments

We define four metrics as proxies for measuring the impact of *defogging* in RTS games, we present some baselines and the results of our models on these metrics, on a human dataset of StarCraft: Brood War games. We use the train, valid, and test set given by (Lin *et al.*, 2017) to do forward modeling and *defogging*. The dataset includes 65k human games of StarCraft: Brood War at generally high skill level. We implemented the models in PyTorch using TorchCraft (Synnaeve *et al.*, 2016).

### 5.1 Metrics

We considered a variety of different metrics to measure the performance of our model on. Two downstream StarCraft tasks obvious for a forward model is strategy prediction and tactics prediction.

In strategy prediction, presence or absence of certain buildings is central to determining what units the opponent will produce. Thus, we can measure the prediction of all opponent buildings in a future frame.

In tactics prediction, the key is to determine the approximate locations of enemy units in the future. Furthermore, we care the most about enemy units that we cannot see. However, it's unclear how to define the fog of war when doing this prediction, since the areas we have vision might change drastically between two skip frames, so we measure the error on both visible and hidden enemy units. Thus, we measure two metrics, one where we define the fog of war to be just where the input frame does not see units, and one where we simply observe all enemy units.

Finally, to use this in control, we wish to minimize the number of miscounts of enemy hidden units.

This results in 4 metrics. All metrics but the first (`g_op_b`) are measured per type per tile. When we measure existence/absence from a regression head, we take a threshold on the regression outputs so that numbers above this threshold counts as existence. These are cross validated per model. The metrics are:

**`g_op_b`** (global opponent buildings) Existence of each opponent building type on any tile, from the classification head.

**`hid_u`** (hidden units) Existence of hidden units, necessarily belonging to your opponent, from the regression head.

**`op_u`** (opponent units) Existence of all opponent units output from the regression head, note that we see some of the opponent units (but they can move or be destroyed).

**`abs_diff`** (absolute difference) An L1 loss on the counts of enemy units, from the regression head, over true positives

| model | metric: grid / step | op_u P | R | F1 | hid_u P | R | F1 | g_op_b P | R | F1 | abs_diff $\sum \|\hat{x} - x\|$ |
|---|---|---|---|---|---|---|---|---|---|---|---|
| conv-lstm2 | 64 / 15 | 0.61 | 0.72 | **0.66** | 0.58 | 0.64 | **0.60** | 0.94 | 0.93 | **0.94** | -3658.00 |
| conv-lstm3 | | 0.61 | 0.71 | **0.66** | 0.58 | 0.63 | **0.60** | 0.94 | 0.93 | 0.93 | -3612.00 |
| striding | | 0.66 | 0.54 | 0.60 | 0.64 | 0.51 | 0.57 | 0.91 | 0.90 | 0.90 | -2412.00 |
| best baseline | | 0.43 | 0.68 | 0.53 | 0.43 | 0.53 | 0.47 | 0.95 | 0.82 | 0.88 | 4580 |
| conv-lstm2 | 32 / 30 | 0.58 | 0.45 | **0.51** | 0.60 | 0.39 | **0.47** | 0.94 | 0.93 | **0.94** | -1126.00 |
| conv-lstm3 | | 0.58 | 0.46 | **0.51** | 0.60 | 0.39 | **0.47** | 0.94 | 0.93 | 0.93 | -1133.00 |
| striding* | | 0.44 | 0.36 | 0.40 | 0.43 | 0.28 | 0.33 | 0.82 | 0.89 | 0.86 | -580.00 |
| best baseline | | 0.32 | 0.34 | 0.33 | 0.21 | 0.32 | 0.26 | 0.95 | 0.81 | 0.88 | 1450 |
| conv-lstm2 | 32 / 15 | 0.60 | 0.48 | **0.53** | 0.61 | 0.42 | **0.50** | 0.95 | 0.93 | **0.94** | -2380.00 |
| conv-lstm3 | | 0.60 | 0.48 | **0.53** | 0.61 | 0.42 | **0.50** | 0.93 | 0.95 | **0.94** | -2382.00 |
| striding | | 0.64 | 0.15 | 0.25 | 0.63 | 0.16 | 0.25 | 0.95 | 0.94 | **0.94** | -498.00 |
| best baseline | | 0.32 | 0.36 | 0.34 | 0.24 | 0.29 | 0.26 | 0.95 | 0.82 | 0.88 | 3050 |
| conv-lstm2 | 32 / 5 | 0.75 | 0.50 | **0.60** | 0.73 | 0.42 | **0.53** | 0.95 | 0.93 | **0.94** | -15713.00 |
| conv-lstm3 | | 0.75 | 0.50 | **0.60** | 0.73 | 0.42 | **0.53** | 0.94 | 0.93 | **0.94** | -15609.00 |
| striding | | 0.62 | 0.47 | 0.54 | 0.60 | 0.40 | 0.48 | 0.92 | 0.91 | 0.92 | -6017.00 |
| best baseline | | 0.49 | 0.40 | 0.44 | 0.26 | 0.51 | 0.34 | 0.95 | 0.83 | 0.89 | 22900 |

Table 1: Score of the models, and of the best baseline for each task (in F1). See Table 2 in Appendix for full details. The absolute difference metric is measured only on the true positives of the model, so we subtract the best baseline to each model score, because a high precision low recall score will have a weaker cost and baseline cost than a high recall model. Thus, as the baseline is different for each model, to give an order of magnitude, we display the lowest (i.e. best of) the best baselines on the eponym line. These `abs_diff` numbers can only compare the models (who beat all the baselines as they are all negatives). More negative is better than baseline. * We could not train a single model to do well on both the regression and classification heads, so we display results for op_b from a striding model with slightly different weights on each head.

## 5.2 BASELINES

To validate the strength of our models, we validate their performance against some hard-coded baselines, similar to what rule-based bots use traditionally in StarCraft: Brood War competitions. These baselines rely exclusively on what was previously seen, as well as some rules of the games (to infer hidden requirements of units that we saw).

We rely on four different baselines to measure success:

1. *Input* – The baseline predicts by copying the input frame.

2. *Perfect memory (PM)* – The baseline remembers everything in the past, and units are never removed, maximizing recall.

3. *Perfect memory + rules (PM+R)* – This baseline is designed to maximize `g_op_b`, by using perfect memory and game rules to infer the existence of unit types that are prequisites for unit types that are seen.

4. *Previous Seen (PS)* – This baseline takes the position of the last seen unit in the map, which is what most rule based bots do in real games. When a location is revealed and no units are at the spot, the count is again reset to 0.

In order to beat these baselines, our models have to learn a good correlation model on the units and buildings type, remember what it has seen before and understand the dynamics of the movement of the units.

## 5.3 RESULTS

We report baselines and models scores according to the metrics described above, on 64 and 32 walktiles effective grids (striding), with time steps of 5, 15, and 30 seconds, in Table 1.

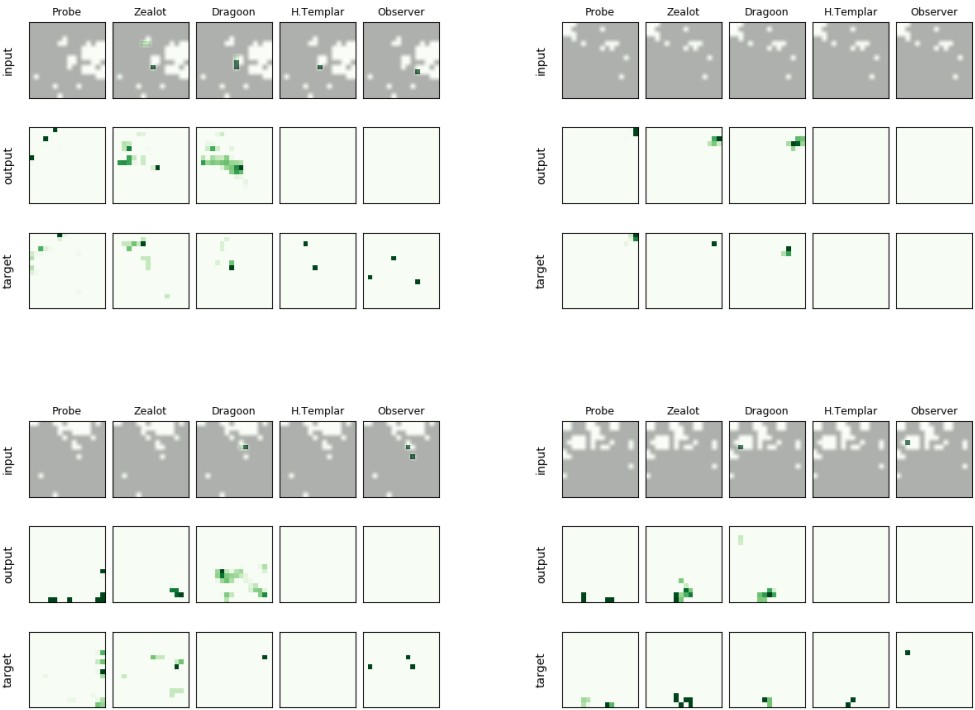

Figure 2: Plots show unit counts of the specified type per map cell (darker dots correspond to higher counts). Top row of each plot shows model inputs (i.e. observed units), middle row shows model predicted unit distributions and bottom row shows real unit distributions. In the left two plots, we observe that our model sees only a few opponent units, and tries to predict the location of their army under the for of war, in places it had not seen units before. In the right two plots, we observe our model remembering the locations of the opponent base and army.

To obtain the existence thresholds from a regression output, we run a sweep on the validation set on 0.1 increments up to 1.5, where the model predicts existence if it predicts more than this value in the output. This value is usually very low, indicating that our model is sure of grid locations with zero units. We also finetune the probability of predicting global opponent building existence the same way, in that we predict existence if the probability output by the model is greater than $p$. Generally, $p$ tends to be slightly above 0.5 to most maximize the F1 score. We report the results on the test set with the best thresholds on the validation set.

We note that for g_op_b prediction, the baselines already do very well. It is hard to beat the best baseline, PM+R. However, most of our models have higher recall than the baseline, indicating that they predict many more unexpected buildings, at the expense of mispredicting existing buildings. We do best above baseline on predicting the global existence of opponent buildings 30 seconds in the future, on whatever grid size.

Our models make the most gains above baseline on unit prediction. Since units often move very erratically, this is difficult for a baseline that only predicts the previous frame. In order to predict units well, the model must have a good understanding of the dynamics of the game as well as the possible strategies taken by players in the game. For our baselines, the more coarse the grid size the easier it is to predict unit movement, since small jitters won't change the featurization. Additionally, predicting closer in time also improves the result, since units move less often.

We do 24% better in F1 in predicting the baseline for hidden opponent units 15 seconds in the future with a grid size of 32. In predicting all opponent units, we do 19% better, since baselines are able to more easily predict the existence of units that are already seen. These are the most useful cases for a *defogger*, since predict tactical movements 15 seconds in the future can aid a bot controller in predicting what opponents will do next. Predicting 5 seconds is not as useful, as enemies will not move much, and predicting at a grid size of 64 is harder for our model to use, as units on the diagonals of a grid box might not even see each other. We notice that our models have much higher precision than the baselines, with some over twice the precision. This supports the observation that our models have a better usage of their memory than the baselines, and are able to remember objects, but also "forget" them based on visibility of their previous positions and other correlates. On all unit prediction tasks, our models beat the baseline by a significant amount.

Finally, we try to give a metric to approximate an estimate of how well the model can do in control. During control, we wish to minimize the number of mispredictions of opponent units. However, we noticed that the absolute number of mispredicted units would average to 10s of thousands per game for the baselines, and only several thousand per game for our model. This is because if we continually mispredict, for example, using the perfect memory baseline, then the mispredictions would add up over the length of the game. This shows how bad the baselines are at the regression task compared to classification, often 10x or more off from our models. To create more sane outputs comparable to the outputs of our models, we only display the L1 score over the true positives of the model.

## 6 CONCLUSIONS

We propose a set of tasks and benchmarks to address partial observability and forward modeling. We provide 4 metrics, predicting enemy units, predicting just hidden units, predicting existence of buildings, and an L1 loss on the number of predictions. Additionally, we provide 4 baselines, Input, Perfect Memory, Perfect Memory + rules, and Previous Seen. These baselines cover a variety of strategies employed by real StarCraft bots.

We made the first attempt of using a large scale deep learning architecture on doing full game forward modeling in StarCraft, a complex environment. We propose a solution based on a hierarchical ConvNet-LSTM encoder-decoder architecture. We used a public dataset of 65k games, to do future frame prediction as well as to recover hidden information from partial observations. We do a large scale model search on the best performing models and hyperparameters, and we find models that beat all baselines on all tasks on all task parameters.

To use a forward model in control, the model should be able to predict the distribution of enemy units at some time in the future. This can be easily used in reactive control. Additionally, the model should be able to predict the existence of key buildings, which can be used in long term planning tasks.

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

## APPENDIX

| model | measure:
grid / step | op_u | | | hid_u | | | g_op_b | | | abs_diff
$\sum \lvert \hat{x} - x \rvert$ |
|---|---|---|---|---|---|---|---|---|---|---|---|
| | | P | R | F1 | P | R | F1 | P | R | F1 | |
| Input | 64 / 15 | 0.74 | 0.27 | 0.40 | 0.00 | 0.00 | 0.00 | 0.97 | 0.25 | 0.39 | |
| PS | | 0.66 | 0.41 | 0.50 | 0.50 | 0.30 | 0.38 | 0.97 | 0.48 | 0.64 | |
| PM. | | 0.43 | 0.68 | 0.53 | 0.43 | 0.53 | 0.47 | 0.94 | 0.70 | 0.80 | |
| PM+R | | 0.25 | 0.70 | 0.37 | 0.20 | 0.57 | 0.30 | 0.95 | 0.82 | 0.88 | |
| conv-lstm2 | | 0.61 | 0.72 | **0.66** | 0.58 | 0.64 | **0.60** | 0.94 | 0.93 | **0.94** | -3658.00 |
| conv-lstm3 | | 0.61 | 0.71 | **0.66** | 0.58 | 0.63 | **0.60** | 0.94 | 0.93 | 0.93 | -3612.00 |
| striding | | 0.66 | 0.54 | 0.60 | 0.64 | 0.51 | 0.57 | 0.91 | 0.90 | 0.90 | -2412.00 |
| input | 32 / 30 | 0.44 | 0.15 | 0.23 | 0.00 | 0.00 | 0.00 | 0.97 | 0.29 | 0.45 | |
| PS | | 0.32 | 0.34 | 0.33 | 0.25 | 0.26 | 0.25 | 0.97 | 0.65 | 0.78 | |
| PM. | | 0.24 | 0.42 | 0.31 | 0.21 | 0.32 | 0.26 | 0.94 | 0.69 | 0.80 | |
| PM+R | | 0.10 | 0.45 | 0.17 | 0.07 | 0.36 | 0.12 | 0.95 | 0.81 | 0.88 | |
| conv-lstm2 | | 0.58 | 0.45 | **0.51** | 0.60 | 0.39 | **0.47** | 0.94 | 0.93 | **0.94** | -1126.00 |
| conv-lstm3 | | 0.58 | 0.46 | **0.51** | 0.60 | 0.39 | **0.47** | 0.94 | 0.93 | 0.93 | -1133.00 |
| striding* | | 0.44 | 0.36 | 0.40 | 0.43 | 0.28 | 0.33 | 0.82 | 0.89 | 0.86 | -580.00 |
| input | 32 / 15 | 0.50 | 0.15 | 0.22 | 0.00 | 0.00 | 0.00 | 0.97 | 0.25 | 0.39 | |
| PS | | 0.32 | 0.36 | 0.34 | 0.24 | 0.29 | 0.26 | 0.97 | 0.64 | 0.77 | |
| PM. | | 0.22 | 0.47 | 0.30 | 0.19 | 0.38 | 0.25 | 0.94 | 0.70 | 0.80 | |
| PM+R | | 0.09 | 0.50 | 0.16 | 0.07 | 0.41 | 0.12 | 0.95 | 0.82 | 0.88 | |
| conv-lstm2 | | 0.60 | 0.48 | **0.53** | 0.61 | 0.42 | **0.50** | 0.95 | 0.93 | **0.94** | -2380.00 |
| conv-lstm3 | | 0.60 | 0.48 | **0.53** | 0.61 | 0.42 | **0.50** | 0.93 | 0.95 | **0.94** | -2382.00 |
| striding | | 0.64 | 0.15 | 0.25 | 0.63 | 0.16 | 0.25 | 0.95 | 0.94 | **0.94** | -498.00 |
| input | 32 / 5 | 0.74 | 0.20 | 0.31 | 0.00 | 0.00 | 0.00 | 0.98 | 0.21 | 0.34 | |
| PS | | 0.49 | 0.40 | 0.44 | 0.36 | 0.30 | 0.33 | 0.97 | 0.58 | 0.72 | |
| PM. | | 0.27 | 0.62 | 0.37 | 0.26 | 0.51 | 0.34 | 0.94 | 0.71 | 0.81 | |
| PM+R | | 0.15 | 0.65 | 0.25 | 0.13 | 0.55 | 0.21 | 0.95 | 0.83 | 0.89 | |
| conv-lstm2 | | 0.75 | 0.50 | **0.60** | 0.73 | 0.42 | **0.53** | 0.95 | 0.93 | **0.94** | -15713.00 |
| conv-lstm3 | | 0.75 | 0.50 | **0.60** | 0.73 | 0.42 | **0.53** | 0.94 | 0.93 | **0.94** | -15609.00 |
| striding | | 0.62 | 0.47 | 0.54 | 0.60 | 0.40 | 0.48 | 0.92 | 0.91 | 0.92 | -6017.00 |

Table 2: Score of the models, and of the best baseline for each task (in F1). The absolute difference metric is measured only on the true positives of the model, so we subtract the best baseline. Thus, the baseline is different for each model, so we do not display it. More negative is better than baseline.

