# OpenReview forum: "Forward Modeling for Partial Observation Strategy Games - A StarCraft Defogger"
_ICLR.cc/2018/Conference — Reject_

### Official Review · AnonReviewer1 · 2017-11-26
**Baselines and metrics for Starcraft defogging**

**Rating:** 5
**Confidence:** 4

**Review:**

The paper considers a problem of predicting hidden information in a poMDP with an application to Starcraft.
Authors propose a number of baseline models as well as metrics to assess the quality of “defogging”.

I find the problem of defogging quite interesting, even though it is a bit too Starcraft-specific some findings could perhaps be translated to other partially observed environments.
Authors use the dataset provided for Starcraft: Brood war by Lin et al, 2017.

My impression about the paper is that even though it touches a very interesting problem, it neither is written well nor it contains much of a novelty in terms of algorithms, methods or network architectures.

Detailed comments:
* Authors should at very least cite (Vinyals et al, 2017) and explain why the environment and the dataset released for Starcraft 2 is less suited than the one provided by Lin et al.
* Problem statement in section 3.1 should certainly be improved. Authors introduce rather heavy notation which is then used in a confusing way. For example, what is the top index in $s_t^{3-p}$ supposed to mean? The notation is not much used after sec. 3.1, for example, figure 1 does not use it.
* A related issue, is that the definition of metrics is very informal and, again, does not use the already defined notation. Including explicit formulas would be very helpful, because, for example, it looks like when reported in table 1 the metrics are spatially averaged, yet I could not find an explicit notion of that.
* Authors seem to only consider deterministic defogging models. However, to me it seems that even in 15 game steps the uncertainty over the hidden state is quite high and thus any deterministic model has a very limited potential in prediction it. At least the concept of stochastic predictions should be discussed
* The rule-based baselines are not described in detail. What does “using game rules to infer the existence of unit types” mean?
* Another detail which I found missing is whether authors use just a screen, a mini-map or both. In the game of Starcraft, only screen contains information about unit-types, but it’s field of view is limited. Hence, it’s unclear to me whether a model should infer hidden information based on just a single screen + minimap observation (or a history of them) or due to how the dataset is constructed, all units are observed without spatial limitations of the screen.

---

### Official Review · AnonReviewer2 · 2017-11-27
**From what I can tell the paper is correct but might lack in novelty or impact**

**Rating:** 4
**Confidence:** 1

**Review:**

The authors introduce the task of "defogging", by which they mean attempting to infer the contents of areas in the game StarCraft hidden by "the fog of war".

The authors train a neural network to solve the defogging task, define several evaluation metrics, and argue that the neural network beats several naive baseline models.

On the positive side, the task is a nice example of reasoning about a complex hidden state space, which is an important problem moving forwards in deep learning.

On the negative side, from what I can tell, the authors don't seem to have introduced any fundamentally new architectural choices in their neural network, so the contribution seems fairly specific to mastering StarCraft, but at the same time, the authors don't evaluate how much their defogger actually contributes to being able to win StarCraft games.  All of their evaluation is based on the accuracy of defogging.

Granted, being able to infer hidden states is of course an important problem, but the authors appear to mainly have applied existing techniques to a benchmark that has minimal practical significance outside of being able to win StarCraft competitions, meaning that, at least as the paper is currently framed, the critical evaluation metric would be showing that a defogger helps to win games.

Two ways I could image the contribution being improved are either highlighting and generalizing novel insights gleaned from the process of building the neural network that could help people build "defoggers" for other domains (and spelling out more explicitly what domains the authors expect their insights to generalize to), or doubling down on the StarCraft application specifically and showing that the defogger helps to win games.  A minimal version of the second modification would be having a bot that has access to a defogger play against a bot that does not have access to one.

All that said, as a paper on an application of deep learning, the paper appears to be solid, and if the area chairs are looking for that sort of contribution, then the work seems acceptable.

Minor points:
- Is there a benefit to having a model that jointly predicts unit presence and count, rather than having two separate models (e.g., one that feeds into the next)?  Could predicting presence or absence separately be a way to encourage sparsity, since absence of a unit is already representable as a count of zero?  The choice to have one model seems especially peculiar given the authors say they couldn't get one set of weights that works for both their classification and regression tasks
- Notation: I believe the space U is never described in the main text. What components precisely does an element of U have?
- The authors say they use gameplay from no later than 11 minutes in the game to avoid the difficulties of increasing variance. How long is a typical game?  Is this a substantial fraction of the time of the games studied?  If it is not, then perhaps the defogger would not help so much at winning.
- The F1 performance increases are somewhat small. The L1 performance gains are bigger, but the authors only compare L1 on true positives. This means they might have very bad error on false positives. (The authors state they are favoring the baseline in this comparison, but it would be nice to have those numbers.)
- I don't understand when the authors say the deep model has better memory than baselines (which includes a perfect memory baseline)

---

> ### Comment · AnonReviewer2 · 2018-01-13
> **Changes during the rebuttal period were not significant**
>
> I appreciate the authors responses to my review, and their emphasis on task definition, but my other main concern about the work (poor evaluation --- no actual gameplay using defogger vs no defogger) remains.  Also, the authors do not mention any added discussion about how to generalize their "defogging" task to other applications, which seems critical to discuss thoroughly if the authors intend introducing the task of "defogging" to be their primary contribution.

---

### Official Review · AnonReviewer4 · 2017-12-17

**Rating:** 5
**Confidence:** 3

**Review:**

# Summary
This paper introduces a new prediction problem where the model should predict the hidden opponent's state as well as the agent's state. This paper presents a neural network architecture which takes the map information and several other features and reconstructs the unit occupancy and count information in the map. The result shows that the proposed method performs better than several hand-designed baselines on two downstream prediction tasks in Starcraft.

[Pros]
- Interesting problem

[Cons]
- The proposed method is not much novel.
- The evaluation is a bit limited to two specific downstream prediction tasks.

# Novelty and Significance
- The problem considered in this paper is interesting.
- The proposed method is not much novel.
- Overall, this paper is too specific to Starcraft domain + particular downstream prediction tasks. It would be much stronger to show the benefit of defogging objective on the actual gameplay rather than prediction tasks. Alternatively, it could be also interesting to consider an RL problem where the agent should reveal the hidden state of the opponent as much/quickly as possible.

# Quality
- The experimental result is not much comprehensive. The proposed method is expected to perform better than hand-designed methods on downstream prediction tasks. It would be better to show an in-depth analysis of the learned model or show more results on different tasks (possibly RL tasks rather than prediction tasks).

# Clarity
- I did not fully understand the learning objective. Does the model try to reconstruct the state of the current time-step or the future? The learning objective is not clearly defined. In Section 4.1, the target x and y have time steps from t1 to t2. What is the range of t1 and t2? If the proposed model is doing future prediction, it would be important to show and discuss long-term prediction results.

---

### Author Response · Authors · 2018-01-05
**Rebuttals**

Thanks for reviewing our paper in details.
We updated the paper with (much) better experimental results after fixing a bug, but no new experiments.
We would like to address the comments by the reviewers.

- Lack of novelty:
This is a task paper, not a model paper. The main contribution is to define the task, baselines, and report on first results (better than baselines). We believe it is sufficient to present a new task. We believe it holds several advantages to forward modeling research on an image domain, such as the environment being closed and having less complexity than the real world.

- Notation:
  - We clarified that U is the set of unit types.
  - s^{(3-p)} means observed state for the player 3 minus the value of p, with p taking values in {1, 2}, which means the opponent of the player who gets to see s^p. We refactored the notation to be more clear.

- About the model:
  - We do have a model that jointly learns to predict presence/absence and unit count, it is the same model that is shared by two different "heads". About the specific "set of weights" footnote, it is for this line only, and it is most likely because of limited hyperparameter sweep.
  - Both the losses are always at t+frame_skip, i.e. t+5/15/30 seconds, as mentioned in the last paragraph of section 4.2.
  - Using a stochastic model defogging model is an interesting extension.
  - We use the full view, but as we pull over number of units per unit types at the input resolution, the input looks more like a minimap with as many channels as unit types.

- About the dataset:
  - length of games: We only go up to 11 minutes, and a median game is approximately this long. A distribution of game length in the dataset can be found in Figure 2 of https://arxiv.org/abs/1708.02139
  - (Vinyals et al. 2017) was not available when we conducted most of the research, this dataset is not less nor more suited than the one provided by Lin et al., it is just additional dataset pre-processing work.

- About results:
  - Taking the L1 norm only over true positives is definitely advantageous to the baselines, without which (meaning, if we include false positives) they get approximately 20 times worse L1 norms, and then the numbers are harder to compare. With the true positive filter, the difference looks like 1000 - 2000 = ~1000, whereas without the filter, it looks like 5000 - 100000 = ~95000.
  - The "much more effective memory model than the baselines" formulation is wrongly worded, we corrected, the difference between the perfect memory (PM, or PM+rules) and previous seen (PS) baselines is what matters: having a perfect memory maximizes the recall, but sometimes it is required to reset the memory when the location is seen. We changed the wording for "This supports the observation that our models have a better usage of their memory than the baselines, and are able to remember objects, but also ``forget'' them based on visibility of their previous positions and other correlates."

---

### Decision · Program_Chairs · 2018-01-29
**ICLR 2018 Conference Acceptance Decision**

**Decision:**

Reject

**Comment:**

The reviewer scores are fairly close, and the comments in their reviews are likewise similar.  All reviewers indicate that they find this to be an interesting learning domain.  However, they also agree in assessing the proposed method as having limited novelty and significance.  They also critiqued the empirical evaluation as being too specific to Starcraft and not comprehensive, without providing evidence that the defogger contributes to winning at StarCraft.  The authors wrote a substantial rebuttal to the reviews, but it did not convince anyone to increase their scores.